# Assessing the Performance of Design Variations of a Thermoacoustic Stirling Engine Combining Laboratory Tests and Model Results

Carmen Iniesta [1,*], José Luis Olazagoitia [1], Jordi Vinolas [1,2] and Jaime Gros [1]

[1] Department of Engineering, Nebrija University, 28015 Madrid, Spain
[2] Escuela Politécnica Superior, Universidad Francisco de Vitoria, 28223 Madrid, Spain
* Correspondence: miniesta@nebrija.es

**Abstract:** The equations governing energy conversion in traveling wave thermoacoustic machines are affected by their multiphysics nature. Their theoretical study is complicated and, in order to obtain real results, it is necessary to resort to prototypes and experimental tests. This work presents the theoretical–experimental study of a thermoacoustic Stirling engine in which, by altering some of its critical parts and analysing the experimental result, it is possible to improve its performance. The methodology used is based on the study and modelling of the active and reactive acoustic power flow for the improvement of the output power of the thermoacoustic engine. The work and analysis are illustrated through the instrumentation of a thermoacoustic Stirling engine with three different feedbacks. The present work presents the experimental results obtained in all cases, including their parameters, experimental data and analysis. The results are compared with the virtual computational models, allowing us to quantify the theoretical/experimental correlation and the performance improvement obtained that allows us to significantly increase the energy provided by the thermoacoustic machine. In conclusion, it is shown that the proposed methodology is a useful design tool that allows using a simplified and practical approach in the study of the power flow of thermoacoustic machines.

**Keywords:** thermoacoustic Stirling engine; reactive acoustic power; acoustic network improvement; early-stage design

## 1. Introduction

The reduction of fuel consumption in the automotive industry has been an important target for many years and has been addressed in different areas. Emerging emissions regulations, the potential for recovery of waste energy in internal combustion engines studies [1] and identification of the energy and environmental implications [2] have given rise to an actual research topic, the adoption of harvesting technologies for the recovery of the wasted heat through the exhaust systems in vehicle internal combustion engines (ICE) [3].

The work presented in this paper is part of a broader research project, RECOVER (Recovery of waste energies from light-duty vehicles. Technological Impact). The main objective of the RECOVER project is to identify the waste energy thresholds of the exhaust gases in up to 90% of the engine operation map (experiment and modelling). The proposal studies processes ranging from the introduction of primary energy (fuel injection into the engine), transformation and use of energy in engine and vehicle (combustion into the cylinders and heat distribution along the exhaust system) as well as the study and develop solutions for recovering thermal and mechanical energy which is wasted in light-duty vehicles and engines. From the point of view of exhaust thermal energy recovery, among the most studied technologies, the most well-known are: ORC (organic Rankine cycles) [4], eTG (electric turbo-generators) [5], TEG (thermoelectric generators) [6] and thermoacoustic Stirling engines [7].

One of the important parameters when trying to recover energy from a heat source is its temperature. In the RECOVER project, the heat source is the exhaust system. The temperature of the exhaust gases very much depends on the engine operating conditions and can range from 400 °C to 650 °C at full load [8]. These conditions offer a good potential for the TA-SLiCE (thermoacoustic Stirling-like cycle engines) to recover waste heat in vehicle exhaust systems [9]. A thermoacoustic exhaust gas recovery solution is visualised in Figure 1.

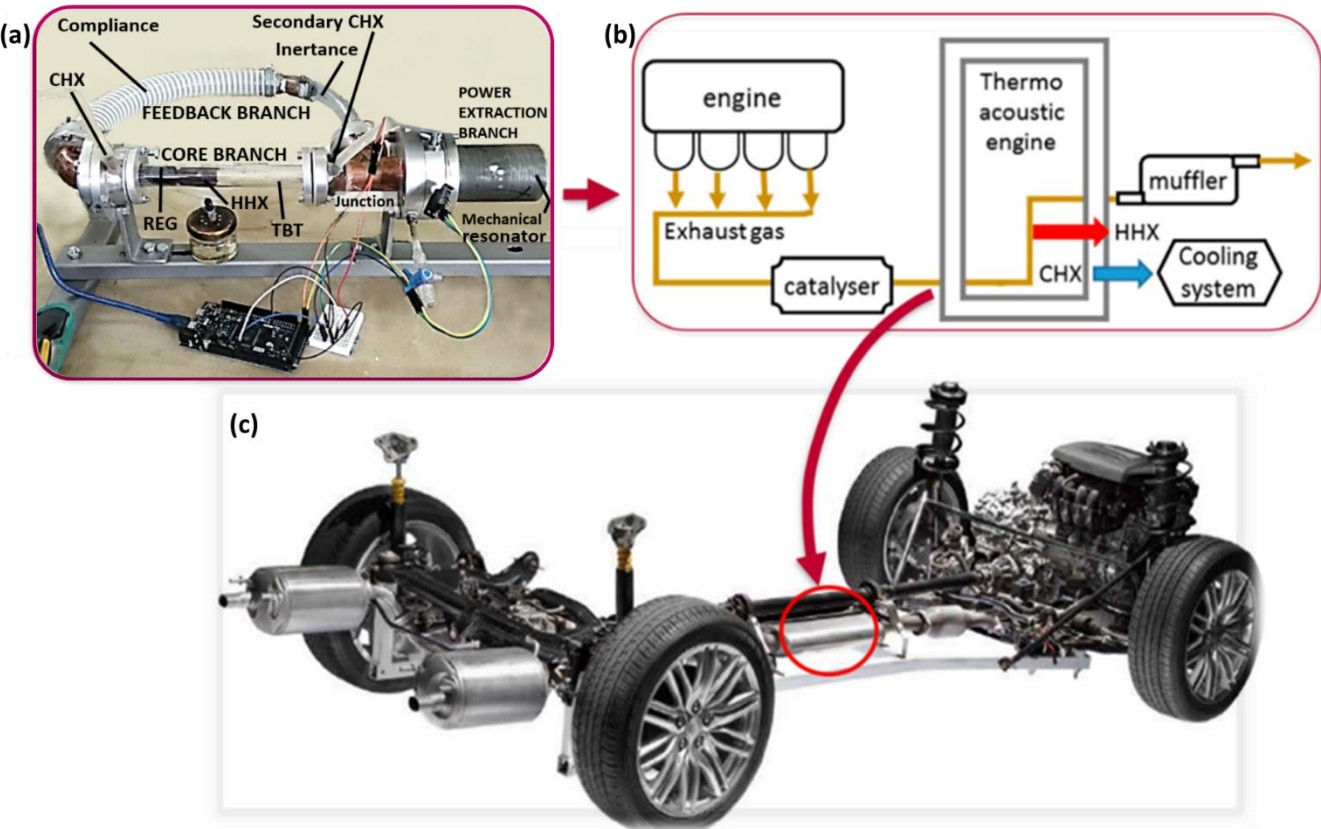

**Figure 1.** (**a**) Picture of the thermoacoustic Stirling engine used in this research. (**b**) Visualisation of a thermoacoustic solution to recover waste heat in exhaust systems. The HHX (hot heat exchanger) of a thermoacoustic engine absorbs heat from the exhaust pipe of a combustion engine and the cold heat exchanger (CHX) releases heat to the refrigeration system of the vehicle itself. (**c**) A possible location of the thermoacoustic engine in the exhaust system.

The development of thermoacoustic energy conversion technology is rather recent. Ceperley presented the first annular travelling-wave engine design in 1979 [10]. Later, Backhaus and Swift achieved a breakthrough in 1999 [11], with the first thermoacoustic Stirling engine. Figure 1a shows the thermoacoustic Stirling engine used in this research. It is a compact design demonstrator of which the main components are: the core branch, which consists of a main and a secondary cold heat exchanger; a TBT (thermal buffer tube); an HHX (high-temperature heat exchanger); and an REG (regenerator). The REG is located between the CHX (main cold heat exchanger) and the HHX (hot heat exchanger). The power extraction branch of the TA-SLiCE shown in Figure 1a is in this case a mechanical resonator which resonates at the same frequency as the gas column in the acoustic network. The passive acoustic ducts and cavities (compliance and inertance in Figure 1a) are also named the feedback branch. The configuration of the passive acoustic circuit (feedback branch) affects the acoustic wave field and therefore the acoustic power flow of the TA-SLiCE generator [12]. There are a variety of feedback arrangements studied to vary the pressure/flow phasing of the working gas flow in a TA-SLiCE [13].

The feedback branch design is critical to the efficient integration of the travelling wave characteristic motion with the Stirling cycle performance. To do this, the DeltaEC (Design Environment for Low-Amplitude Thermoacoustic Energy Conversion) software developed in Los Alamos National Laboratory [14] is widely used, and it represents a significant advance in the development of thermoacoustic devices. The conventional energy analysis of thermoacoustic devices based on DeltaEC models considers that inertance (L) and compliance (C) in these systems do not dissipate or produce acoustic energy. They simply transmit it by modifying the pressure phase $p_1$ or the flow rate $U_1$ [15]. The acoustic power dissipation included in the DeltaEC calculation of any duct section is only due to resistance. The inertance is generally accompanied by viscous resistance R, and compliance is generally accompanied by thermal-relaxation resistance R, with both resistances dissipating acoustic power.

Throughout the last few years, an increasing number of acoustic phase-adjustment concepts have been proposed for the travelling-wave thermoacoustic engines based on the linear thermoacoustic theory. Several options have been used to "adjust" the acoustic field to adapt the velocity of the gas along the regenerator's temperature gradient nearly in phase with the oscillating pressure, among others:

- A side branch stub introduced to correct the acoustic field within the looped-tube engine altered by the installation of a linear alternator [16].
- A side-branched Helmholtz resonator to control and tune the phase angle between the velocity and pressure amplitude in looped-tube traveling-wave thermoacoustic engine [17].
- A compliance and an inertance tube are proposed as the phase adjustor to control the acoustic field in a looped-type thermoacoustic engine [18].
- The introduction of a phase modulation object (ellipse) at the acoustic power output port to adjust the acoustic field distribution and improve impedance matching [19].
- An in-line piston located inside the main resonator introduced in order to optimise the acoustic field from any disturbances [20].
- A compliance serves as a tuner for the phase difference between the volume flow rate and the pressure oscillation in the core section of a closed-loop system [21].

The solutions proposed above are based on complex mathematical background analysis of the active acoustic power flows propagating in the engine. However, an energetic analysis methodology, the RAP (reactive acoustic power-flow) methodology, was recently published [22]. Avoiding the mathematical complexity, this new methodology leads to a substantial simplification and comprehension of the energetic analysis of the TA-SLiCE by combining active and reactive acoustic power flow calculations in order to assess the influence of the impedances of each component of the acoustic circuit in the pressure $p_1$ or flow rate $U_1$ phasors [23–25]. There is no published paper in which a real TA-SLiCE has been studied via the RAP methodology. According to this topic, a fully energetic model for the engine is presented. Thereafter, engine specifications such as active and reactive acoustic power flows can be estimated with this model. Finally, the RAP methodology is validated using a laboratory demonstrator.

As mentioned before, the present work intends to investigate the energetic behaviour of a TA-SLiCE demonstrator (with low operating frequency and pressure) using the RAP methodology for the first time. Indeed, this approach based on the energetic analysis performed by the RAP methodology, shows a performance improvement of a real thermoacoustic Stirling engine demonstrator when reactive acoustic power is included as a critical design parameter. Only compliance feedback design modifications are made during the experiments. In this regard, the variations of the experimental power measured in the power extraction branch infer that, in the regenerator, they are only due to the reactive power produced by the compliance. Hence the RAP methodology is experimentally validated in this paper, for the first time, where the active and reactive acoustic power flows propagating in the TA-SLiCE are determined with DeltaEC.

The main challenge of this paper is not focused, as the above literature is, on discussing the concept and mechanism of wave adjustment by considering the feedback only as a phase adjustor. In this research, the RAP methodology, which is based on the feedback compliance as a generator and distributor of reactive acoustic power, is investigated and experimentally validated. The results obtained experimentally demonstrate that this methodology can be effectively used to the design procedure of the ideal acoustic field in a TA-SLiCE, which also allows us to simplify energy analysis of the whole engine by simply adding algebraically the active and reactive acoustic power flows.

The rest of the manuscript is structured as follows: Section 2 describes the conceptual TA-SLiCE design and summarises the results of the modelling results. Section 3 includes the experimental demonstrator assembly presentation and the tests carried out. Section 4 gives the results discussion, while the most relevant conclusions are condensed in Section 5.

## 2. Demonstrator Computational Model

This section presents the conceptual design of the loop-branched-type TA-SLiCE demonstrator that is used for the simulation and tests. The design has three variants (Figure 2), where all of them have the same core and power extraction branches but incorporate three different feedback branches, as Figure 2a–c illustrates. Starting from "Fba" (Feedback branch a), the feedback branches "Fbb" (Feedback branch b) and "Fbc" (Feedback branch b) have a higher compliance volume (in an AC electric circuit analogy this would imply an increase in condenser capacity).

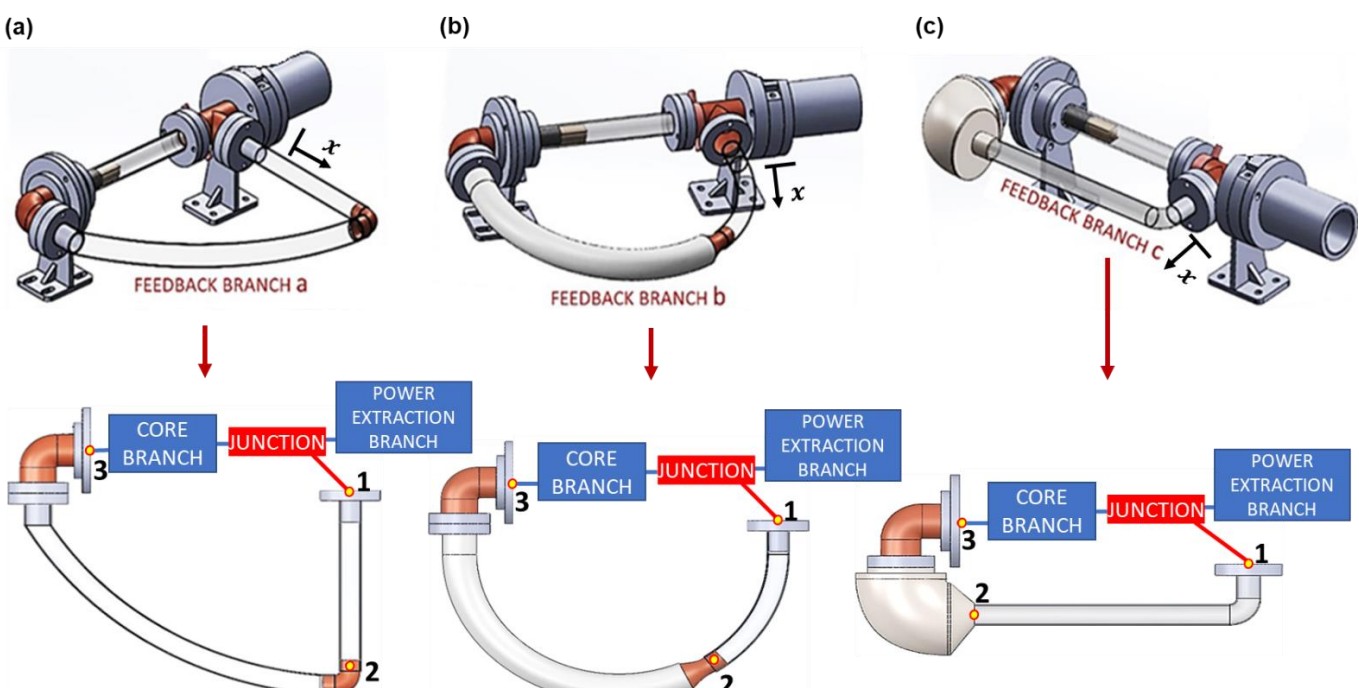

**Figure 2.** Schematic of the three feedback variants of the demonstrator: (**a**) initial design "Fba"; (**b**) first variation design "Fbb"; (**c**) second variation design "Fbc".

The reason behind the design of the three variants is twofold: on the one hand, to carry out a theoretical–experimental correlation of the demonstrator performance and, on the other hand, to illustrate the advantages of the RAP methodology [22].

The RAP methodology allows the design and energetic analysis of functional TA-SLiCEs without the use of phasors. It uses the same theoretical foundations widely used in the study and design of AC (alternating current) electrical circuits. All the dimensions in the three variants shown in Figure 2 are shorter than the acoustic wavelength, so the lumped-impedance approximation based on the electroacoustic analogy, explained in

detail by Swift [15], are directly applicable. In Figure 3, the impedance diagram for the demonstrator feedback branch design is shown.

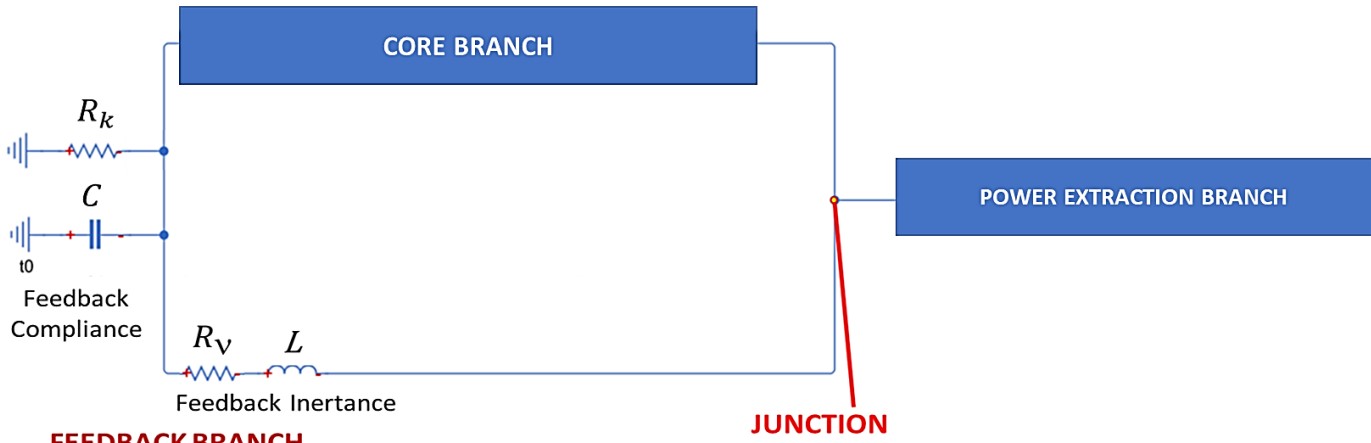

**Figure 3.** Schematic impedance diagram for the demonstrator feedback branch design showing inertance $L$ in series with a compliance $C$. The viscous resistance $R_v$ in series with $L$ and the thermal-relaxation resistance $R_k$ in parallel with $C$.

The feedback for each variant is approximately modelled as an inertance $L$ (this corresponds to Section 1 to 2 of Figure 2a–c) in series with a compliance $C$ (this corresponds to Section 2 to 3 of Figure 2a–c). The resistances associated with these two components are also shown in Figure 3. Hence, the impedance diagram of Figure 3 symbolically represents the most important dynamic features of the demonstrator feedbacks in Figure 2.

For a conventional electro-acoustic analogy, the acoustic impedance $(Z_a)$ is analogous to the impedance in an electric circuit [26]:

$$Z_a = p_1/U_1 = R_a + j(\omega L_a - 1/\omega C_a) \quad \leftrightarrow \quad Z = V_1/I_1 \qquad (1)$$

where $\omega$, $p_1$, $U_1$ are the angular frequency, the pressure amplitude and the volumetric velocity, respectively. $R_a$, $L_a = \rho_m \Delta x / A$ and $C_a = V/\gamma p_m$ are the acoustic resistance, the inertance and the compliance, respectively. $\rho_m$, $p_m$, $A$, $\gamma$, $\Delta x$ and $V$ are mean density, mean pressure cross-sectional area, specific heat ratio, inertance duct length and compliance volume, respectively.

In electrical engineering, the PF (power factor) of an AC power system is the ratio of active power $(P_A)$ to the total power (apparent power, $P_Z$). Based on the electroacoustic analogy, the total power of the inductance $P_Z = V_1 \times I_1$ is a phasor, which is also known as apparent power. $P_Z$ can be divided into active power, $P_A$, and into reactive power, $P_R$.

The DeltaEC models developed for the three prototype variants calculate these two powers numerically in every location of the system [22].

The real part of the phasor $P_Z$ is analogous to the active acoustic power flow. It represents the usable power since it is the only one that is transformed into work. This variable averaged over an integral number of cycles, circulating in the positive $x$ direction through a duct is given according to the Rott linear approximation [15] by the second order expression:

$$\dot{E}(x) = \frac{\omega}{2\pi} \oint pU dt = Re\left[p_1(x)e^{i\omega t}\right] Re\left[U_1(x)e^{i\omega t}\right] dt = \frac{1}{2} Re(\tilde{p}_1 U_1) = \frac{1}{2} Re\left(p_1 \tilde{U}_1\right) = \frac{1}{2}|p_1||U_1| cos\varnothing_{pU} \qquad (2)$$

The imaginary part of the $P_Z$ phasor is analogous to the reactive acoustic power flow. It represents the power consumed by all the devices that have some type of coil or inductance;

it overloads any circuit, increasing the current (the volumetric velocity of the acoustic wave, by electro-acoustic analogy) that circulates through it, without producing work:

$$\dot{Q}(x) = \frac{1}{2} Im(\widetilde{p}_1 U_1) = \frac{1}{2}|p_1||U_1|sin\varnothing_{pU} \tag{3}$$

where $\varnothing_{pU}$ is the phase lead of $p_1$ relative to $U_1$.

The aim of the resonant network in a TA-SLiCE is to adapt $\varnothing_{pU}$, allowing the acoustic wave to reach the regenerator in the best travelling-wave conditions so that the thermodynamic cycle (Stirling type) is carried out in the most efficient way. Thus, the active acoustic power represents the power in a pure travelling acoustic wave, for which $\varnothing_{pU} = 0$ and the reactive acoustic power represents the power in a pure standing wave, for which $\varnothing_{pU} = \pi/2$. The apparent power always has a real and an imaginary contribution so that both active and reactive power circulation are the performance indicators to calculate with DeltaEC, according to the RAP methodology.

*Energetic Analysis Based on the RAP Methodology*

This section intends to study the apparent power flow propagating in the TA-SLiCE according to the RAP methodology. To this aim, the data of the laboratory demonstrator developed by the authors of this paper is used for this assessment. Table 1 shows the main specifications of the engine.

**Table 1.** Known operating parameters of the TA-SLiCE.

| Parameter | Value |
|---|---|
| Operating frequency ($f$) | 22.7 Hz |
| Mean pressure ($P_m$) | $1 \times 10^5$ Pa |
| Thermal input power ($\dot{Q}_H$) | 40 W |
| Hot air temperature ($T_H$) | 429 °C |
| Cold air temperature ($T_C$) | 30 °C |

To assess the performance of the engine, the reactive acoustic power generated by the compliance of the three feedback variants should be addressed from the following three DeltaEC models, where only the section area and length of the compliance are modified according to two objective requirements: the reactive acoustic power must decrease while the active acoustic power must increase or at least not decrease significantly.

- Reference model "Fba" for the feedback design, shown in Figure 2a: DeltaEC multi-parametric fitting method satisfies a variety of mixed boundary conditions and allows the user choosing which variables are calculated as outputs. Therefore, active and reactive acoustic power are numerically calculated at any particular location of the engine.
- Modified model "Fbb", shown in Figure 2b: by increasing the compliance inner diameter to 25 mm and shortening its length to 33 cm. The procedure is the same, and DeltaEC calculates the active and reactive acoustic power along with the rest of the parameters indicated for "Fba".
- Modified model "Fbc", shown in Figure 2c: by increasing again the compliance inner diameter to 63 mm and shortening its length to 12 cm; alike the previous two models, DeltaEC provides again the new active and reactive acoustic power, along with the rest of the parameters indicated above.

Based on the descriptions provided in this section, the simulated DeltaEC outcomes can be abstracted in Table 2. The amplification of the active power by the core branch, $\dot{E}_{Ampli}$, is the difference between the active power calculated after the secondary CHX and before the main CHX. The active acoustic power drop through the feedback branch, $\dot{E}_{Loss}$, is the difference between the active power calculated after the compliance and before the

inertance. The active acoustic power which is not derived to the feedback branch, $\dot{E}_{Extract}$, is a useful acoustic power delivered to the power extraction branch. It is the difference between the active power calculated after the secondary CHX and before the inertance. The reactive acoustic power supplied to the core branch, $\dot{Q}_{Core}$, is calculated before the main CHX.

**Table 2.** Results for the power flows of the three DeltaEC feedback branch variants.

| Calculated Parameters | "Fba" | "Fbb" | "Fbc" |
|---|---|---|---|
| Reactive acoustic power supplied to the core branch, $\dot{Q}_{Core}$ (VAr) | 1.408 | 1.104 | 0.855 |
| Active acoustic power loss through the feedback branch, $\dot{E}_{Loss}$ (W) | 0.185 | 0.157 | 0.255 |
| Amplification of the active acoustic power through the core branch, $\dot{E}_{Ampli}$ (W) | 1.672 | 1.721 | 2.482 |
| Acoustic power delivered to the power extraction branch, $\dot{E}_{Extract}$ (W) | 1.487 | 1.564 | 2.227 |

Table 2 shows that the feedback branch "Fbc" has the higher active acoustic power loss through the feedback branch, $\dot{E}_{Loss}$ and, however, provides the nearest travelling-wave acoustic wave to the core branch rather than the other feedback branches, given that $\dot{Q}_{Core}(Fba) > \dot{Q}_{Core}(Fbb) > \dot{Q}_3(Fbc)$. Therefore, the $\dot{E}_{Ampli}$ and the $\dot{E}_{Extract}$ achieved for the "Fbc" model are the best possible for the given TA-SLiCE demonstrator. In addition, the use of the RAP methodology results in the application of an increase in the inner diameter and a decrease in the length of the compliance. The results in Table 2 show a significant improvement of 50% in the active acoustic power delivered to the extraction branch, $\dot{E}_{Extract}$, due to the modifications proposed by the RAP methodology. Additionally, the better numerical thermoacoustic efficiency achieved for the "Fbc" is:

$$\eta_{ta} = \frac{\dot{E}_{Extract}}{\dot{Q}_H} \approx 5.6\% \tag{4}$$

Here, efficiency is the acoustic power delivered into the resonator to the right of the junction labelled in Figure 3, divided by the electric power supplied to the TA-SLiCE's heater, $\dot{Q}_H = 40$ W. This thermoacoustic efficiency improves significantly the values reported in the literature for similar tiny TA-SLiCEs [13].

### 3. Laboratory Demonstrator

*3.1. Apparatus Assembly*

Based on the above RAP methodology, a TA-SLiCE demonstrator and its three variants is developed. Figure 4 shows the photograph of the experimental demonstrator's branches. These consist of a core branch (a heater, a regenerator, a thermal buffer tube and two cold heat exchangers), a power extraction branch (a rigid moving piston and linear mechanical resonator) and a feedback branch (a loop-branched resonator, consisting of an inertance and compliance). Dimensions of these main components are listed in Tables 3–5.

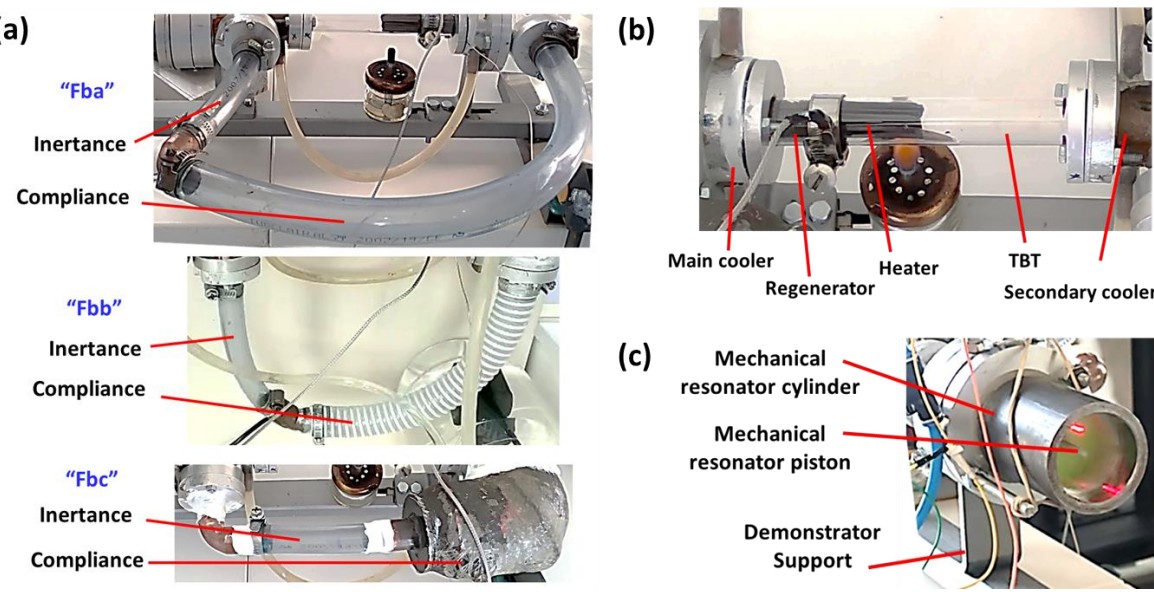

**Figure 4.** TA-SLiCE demonstrator's feedback branch assemblies (**a**), core branch assembly (**b**) and power extraction branch assembly (**c**).

**Table 3.** Main parameters of the TA-SLiCE demonstrator's feedback branch assemblies.

| Parameters | Values |
|---|---|
| Feedback Branch "a" ("Fba") | |
| Compliance: Internal diameter ($D_{Ca}$) | 0.02 m |
| Length ($L_{Ca}$) | 0.52 m |
| Volume ($V_{Ca}$) | $163 \times 10^{-5}$ m$^3$ |
| Inertance: Internal diameter ($D_{La}$) | 0.015 m |
| Length ($L_{La}$) | 0.18 m |
| Feedback Branch "b" ("Fbb") | |
| Compliance: Internal diameter ($D_{Cb}$) | 0.025 m |
| Length ($L_{Cb}$) | 0.33 m |
| Volume ($V_{Cb}$) | $13.4 \times 10^{-5}$ m$^3$ |
| Inertance: Internal diameter ($D_{Lb}$) | 0.015 m |
| Length ($L_{Lb}$) | 0.15 m |
| Feedback Branch "c" ("Fbc") | |
| Compliance: Internal diameter ($D_{Cc}$) | 0.063 m |
| Length ($L_{Cc}$) | 0.12 m |
| Volume ($V_{Cc}$) | $37.4 \times 10^{-5}$ m$^3$ |
| Inertance: Internal diameter ($D_{Lc}$) | 0.015 m |
| Length ($L_{Lc}$) | 0.19 m |

In this demonstrator, the atmospheric air is employed as the working gas. This allows the selection of low-cost plastic tubes as inertance and compliance components, shown in Figure 4a, to simplify the assembling and dismantling of the three feedback branch variants studied in this paper. In the three feedbacks listed in Table 3, an internal diameter was chosen for the inertance and compliance tubes following a single criterion based on commercial availability, except for the compliance "Fbc", shown in Figure 4b, made with 3D-additive manufacturing technology.

**Table 4.** Main parameters of the TA-SLiCE demonstrator's core branch assembly.

| Parameters | | Values |
|---|---|---|
| Core branch | | |
| Main cooler: | Length ($L_{CHX}$) | 0.013 m |
| | Porosity ($\phi_{CHX}$) | 98% |
| Regenerator: | Length ($L_{REG}$) | 0.025 m |
| | Porosity ($\phi_{REG}$) | 75% |
| | Hydraulic radius ($r_{hREG}$) | 0.00033 m |
| Heater: | Length ($L_{HHX}$) | 0.026 m |
| | Porosity ($\phi_{HHX}$) | 91.6% |
| | Thermal input power ($\dot{Q}_H$) | 40 W |
| TBT: | Length ($L_{TBT}$) | 0.085 m |
| | Cross $-$ sectional area ($A_{TBT}$) | 0.00025 m$^2$ |
| Secondary cooler: | Length ($L_{SCHX}$) | 0.013 m |
| | Porosity ($\phi_{SCHX}$) | 100% |

**Table 5.** Main parameters of the TA-SLiCE demonstrator's power extraction branch assembly.

| Parameters | Values |
|---|---|
| Power Extraction Branch | |
| Resonator Cylinder Internal Diameter ($D_{RES}$) | 0.04 m |
| Length of the resonator cylinder ($L_{RES}$) | 0.085 m |
| Length of the power extraction branch ($L_{PEB}$) | 0.135 m |
| Piston mass ($M_{piston}$) | 0.023 kg |
| Piston length ($L_{piston}$) | 0.02 m |

In this section it is worth noting that the present paper is focused on the analysis and performance assessment of the demonstrator based on the feedback branch acoustic design by the RAP methodology. Hence, it is necessary to point out that both the core branch and the power extraction branch have remained invariant in all the experiments performed in this research. Therefore, the qualitative estimations of the values for the heat power in and out of the demonstrator through the hot and cold heat exchangers do not affect the scope nor the result of this study.

Regarding the core branch components listed in Table 4 and shown in Figure 4b, the regenerator is arranged between the main cold and hot heat exchangers. The main and secondary cold heat exchangers are made of copper and aluminium. Both, as shown in Figure 4b, are connected to a water-cooling circuit by means of specifically manufactured jackets inside the coolers. Qualitatively, it was estimated that the thermal power absorbed by the refrigerators is $\dot{Q}_{mainr} \approx 27$ W and $\dot{Q}_{secondaryr} \approx 13$ W. The heat absorbed by the hot heat exchanger is $\dot{Q}_{in} \approx 40$ W, also qualitatively obtained from DeltaEC simulations. The hot heat exchanger, the heater labelled in Figure 4b, supplies that $\dot{Q}_{in}$ to the hot end of the regenerator from the heat source. In the conceptual design of this paper, the heater is located inside a 17.7 mm diameter Pyrex tube, and the heat source is an alcohol burner placed below the HHX to achieve heat transfer from the burning flame. Based on the physical properties of borosilicate glass, the maximum temperature to which it does not deform, or melt, is 450 °C, according to the manufacturer. Therefore, the temperature in the hot side of the regenerator is 450 °C, according to the DeltaEC simulations. This is around the 25% of the thermal power provided by the alcohol burner used as the thermal energy source.

It is well known that the power density in thermoacoustic devices is directly proportional to the mean pressure ($p_m$), and a high $p_m$ in the device results in high acoustic power [15]. As previously mentioned, the gas inside the demonstrator is air at atmospheric pressure. However, increasing the $p_m$ above the atmospheric pressure leads to design complications and construction limitations that result in increased safety and costs. An

example of increased design complexity is the result of the inverse relationship between thermal penetration depth ($\delta_k$) with the mean pressure. A high $p_m$ is translated into a small $\delta_k$, which in turn means that the regenerator has very small ducts, which makes its manufacturing more complicated. The regenerator used in this engine is made of stacked stainless-steel screen discs, with number 40 mesh, made with a wire diameter of 0.44 mm, due to commercial availability. This type of regenerator allows perfect thermal contact conditions between the working gas and its solid surface. The acoustic power that it generates is proportional to its length, but it has the drawback of increasing the losses due to viscosity. Therefore, the design of this critical component requires a compromise between acoustic power generation and acoustic power losses.

In the power extraction branch to the right of the junction labelled in Figures 1–3 of the engines, a moving piston convert the acoustic energy into mechanical energy. The position for the power extraction branch is about 50 mm away from the secondary cooler in the positive direction of wave propagation, at the end of the core branch. Both branches are connected by a copper T-junction. The arrangement of the power extraction branch assembly is specified in Table 5 and explained in the following paragraphs.

It is necessary to take into account that, as shown in Figure 4c, the back of the resonator cylinder was opened to air through the gap between the piston and the resonator cylinder liner that contains it. The machining of the aluminium free piston used in the experiments was carried out on a manual lathe and has a radial clearance, depending on the cylinder liner, of about 10 μm.

### 3.2. Power Extraction Branch Measurements

This section describes in detail the experimental procedure to measure the work performed by the piston for one of the feedback branch variants as the same applies for the other two.

Based on the above demonstrator set-up, work can be exchanged with the external surroundings only in the piston. The active acoustic power flow in the power extraction branch is the time-averaged mechanical power (which is exactly the same to the acoustic power flowing from the face of the piston into the gas).

When the response of the TA-SLiCE demonstrator reaches a steady state of oscillation, the pressure ahead of the piston and piston displacement data are recorded simultaneously by pressure and displacement sensors (Figure 5). The volume change due to piston motion is calculated theoretically from the piston area and piston displacement. However, it should be noted that the mechanical circumferential fit of the pistons over their entire longitudinal stroke is not perfect, and a compromise needs to be reached between friction and air leakage. In short, the fit between piston and cylinder liner has to be machined to facilitate the movement of the pistons over their entire stroke with low friction but, at the same time, to avoid circumferential air leakage to the outside as much as possible. This is therefore a complicated mechanical compromise. In the manufactured demonstrator there are therefore air leaks to the outside that cause the theoretically calculated effective volume to have a deviation. Due to the difficulty in considering the leakage, this issue is often theoretically disregarded. However, in this study, this effect has been taken into account by means of a coefficient accounting for volume reduction. This coefficient is a variable value depending on the particular design. In the case of the present study, the coefficient has been calculated to range from $\mu = 0.25$ for "Fbb" and "Fbc" to $\mu = 0.5$, according to DeltaEC simulations.

The mechanical resonator cylinder contains the piston oscillating between two gas springs, in sliding contact with the cylinder. The pressure at the power extraction branch was measured with a NXP piezoresistive differential pressure transducer (MPX5050DP). The piston position was measured by an OPTIMESS MC-OMS 4140 laser distance sensor. Both data sets were registered using a Brüel & Kjaer data acquisition system (hardware) which was used in combination with the the RT Pro Photon software (7.20, Brüel & Kjaer Ibérica, Madrid, Spain). Then, the experimental measurements of each magnitude of the TA-SLiCE demonstrator were analysed, and the calculation of the acoustic power flow

delivered towards the power extraction branch was also performed. The experimental results are finally compared with the simulated results in Section 4.

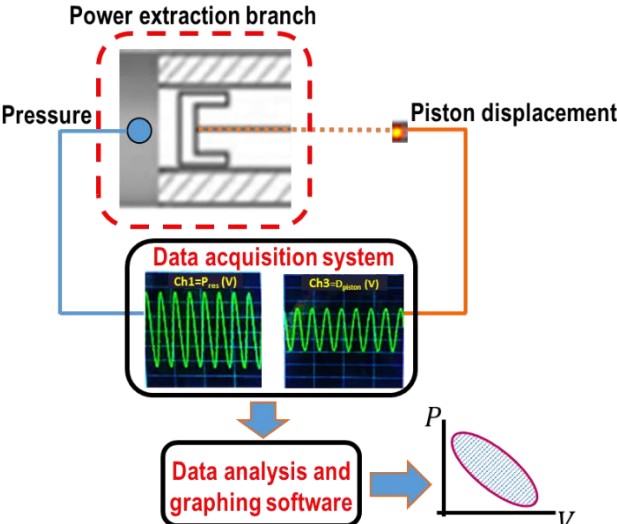

**Figure 5.** Schematic of the experimental data acquisition and analysis process. The acoustic power delivered in front of the piston was obtained experimentally from the area enclosed in the *pV* diagrams obtained from pressure and piston displacement sensors.

The calculation of the acoustic power dissipated in the power extraction branch of the "Fbc" is carried out as follows. V is the volume of oscillating gas in front of the piston, and hence U = dV/dt the volumetric velocity. The time-averaged acoustic power was obtained by using Equation (5). The operating frequency $\omega$, the phase angle $\varnothing_{px}$ between the wave pressure and the position of the resonator piston signals, the acoustic wave pressure amplitude $|p_1|$ and the position amplitude of the piston $|x_1|$, were experimentally obtained from the engine-transducer assembly resonating at $f = 22.7$ Hz. The acoustic wave pressure and the piston position were well approximated by pure sinusoids with a pressure amplitude and piston position amplitude of 8 kPa and 20 mm, respectively, evaluated in an average acoustic period.

To make the acoustic power measurements representative of the data registered during the acquisition time, the same procedure was performed in 10 different cycles over 5 s. Finally, the amplitude of both signals was obtained, $|p_{c1}|$ = (7.9 ± 0.3) kPa and $|x_{c1}|$ = 18.8 ± 0.8 mm, where the indicated error was the standard deviation obtained from the statistics when averaging 11 cycles. Since both signals have been measured simultaneously with the same time base, they can be plotted together in Figure 6, so that the phase angle between the two signals can be obtained directly from the difference in the peak value of both signals. The average phase angle obtained is $\varnothing_{px}$ = 38.6 ± 3.6°.

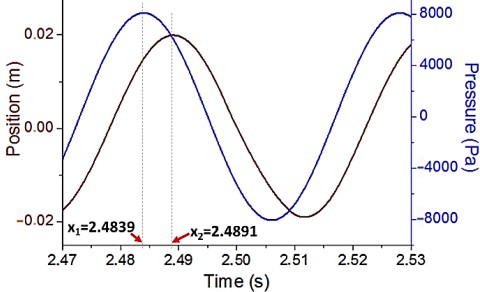

**Figure 6.** Representation of the acoustic wave pressure and the position of the resonator piston signals, acquired with the same time base for an acoustic cycle.

The experimental acoustic power delivered to the power extraction branch can be obtained using the $|p_1|$, $|x_1|$ and $\varnothing_{px}$ acquired data [15]:

$$\dot{E}(x) = \frac{\mu A}{2} Re\left(p_1 \tilde{U}_1\right) = -\frac{\omega \mu A}{2} Im(p_1 \tilde{x}_1) = -\frac{\omega \mu A}{2}|p_1||x_1|sin\varnothing_{px} = -f\pi\mu A|p_1||x_1|sin\varnothing_{px} = -2.06 \pm 0.08\ W \quad (5)$$

where $A$ is the piston area on which the pressure acts; $\omega$, $f$ are the angular and operating frequency of the system, respectively; and the tilde in $\tilde{U}_1$ and $\tilde{x}_1$ denotes complex conjugation. The angle $\varnothing_{px} = \varnothing_{pU} - \frac{\pi}{2}$ is the phase difference by which lead of pressure to the displacement.

An alternative to obtain the acoustic power in a period of piston oscillation, is to calculate the work that the acoustic wave exerts on the piston in that period of oscillation. This was performed by selecting a time interval corresponding to the same period of piston oscillation in the pressure and position signals. That work, performed in one acoustic cycle by the working fluid on the piston, was obtained experimentally from the area enclosed in the ellipse $p_1 V_1$, as shown in Figure 7:

$$\oint p_{c1}\mu A dx = \oint p\mu dV = -0.089\ J$$

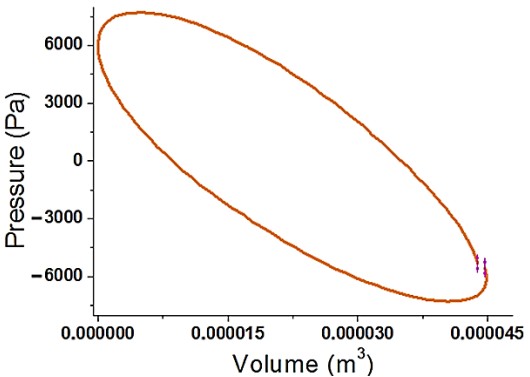

**Figure 7.** Pressure-volume diagram ($p_1 V_1$), where the area enclosed by the ellipse is the net work performed in an acoustic cycle by the working fluid on the resonator's piston.

The experimental acoustic power of an average working cycle was obtained from the area of the ellipse shown in the $p_1 V_1$ diagram of Figure 7:

$$\dot{E}(x_{c1}) = \frac{\omega}{2\pi} \oint p\mu U dt = \frac{\omega}{2\pi} \oint p\mu dV = f \oint p_{c1}\mu dV = -2.03 \pm 0.15\ W$$

This power obtained from the area enclosed in that diagram agrees by 98% with the power given by Equation (5). The negative sign of the power indicates that it was dissipated power.

## 4. Results and Discussion

This section compares experimental data results to the DeltaEC simulated results.

Figure 8 shows the $p_1 V_1$ diagrams obtained from pressure and piston displacement sensors for the "Fba", "Fbb" and "Fbc".

Thus, the difference in the acoustic power delivered towards the power extraction branch can be clearly observed as a function of the feedback branch installed in the demonstrator. Based on the descriptions provided in this section, the experimental outcomes can be abstracted in Table 6.

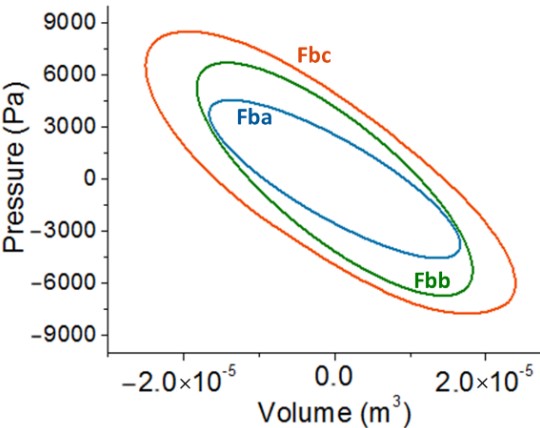

**Figure 8.** Comparison of the pressure–volume diagram ($p_1 V_1$) obtained for the three feedback branches during an average acoustic cycle.

**Table 6.** Comparison of the numerical and experimental results obtained for each variant of the TA-SLiCE demonstrator feedback branch under study.

| Acoustic Power Delivered towards the Power Extraction Branch | Simulation Result (W) | Experimental Result (W) | Error (%) |
|---|---|---|---|
| Feedback "Fba" $\dot{Q}_{Core} = 1.408$ (VAr) | 1.49 | 1.37 | 8.8 |
| Feedback "Fbb" $\dot{Q}_{Core} = 1.104$ (VAr) | 1.56 | 1.41 | 10.6 |
| Feedback "Fbc" $\dot{Q}_{Core} = 0.855$ (VAr) | 2.23 | 2.03 | 9.9 |

Table 6 demonstrates that the RAP methodology can be effectively used to the design procedure of the ideal feedback branch in a TA-SLiCE when reactive acoustic power is included as a critical design parameter.

Based on the experimental data, the history of piston displacement during 108 acoustic cycles is displayed in Figure 9.

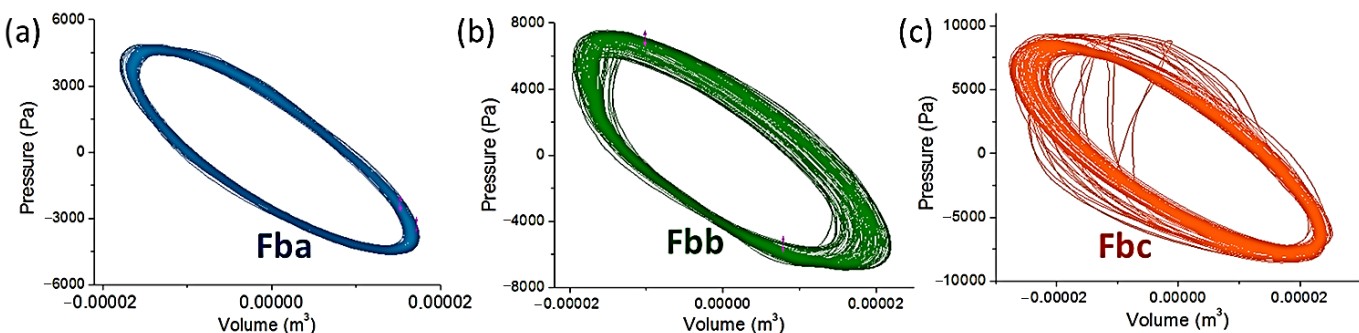

**Figure 9.** Comparison of the pressure–volume diagram ($p_1 V_1$) obtained during the 108 acoustic cycles, with acquisition time of 5 s for: (**a**) feedback branch a, (**b**) feedback branch b and, (**c**) feedback branch c.

The piston displacement becomes increasingly out of control when more energy is delivered to the power extraction branch, as can be seen in Figure 9. This fact introduces losses not properly taken into account by the model. Figure 9c shows how the impact of the piston with the inner wall of the resonator cylinder is affected. Similar phenomena can also be observed, although less pronounced in Figure 9b. Finally, in Figure 9a the phenomenon is practically not observed. This experimental observation is in agreement with the applied de-rating coefficient μ, since for the "Fbb" and "Fbc" cases the de-rating

coefficient is μ = 0.25. However, for the "Fba" case, the applied de-rating coefficient is μ = 0.5, which indicates that the working gas blow-by leakage that occurs is lower, given a stable oscillation in the dynamic response of the engine.

## 5. Conclusions and Future Work

In this paper, the RAP methodology in the loop-branched TA-SLiCE systems has been analysed and demonstrated for the first time in a real demonstrator designed to be able to change its acoustic feedback branch. Three design variants of the same engine have been evaluated and the influence of the acoustic feedback has been assessed. Testing set-up and experimental data processing has been provided to explain the differences in the acoustic power at the power extraction branch. Experimental data have been completed with data from the virtual models of the three demonstrator variants. The models provide extra information and a better understanding of the prototype energetic power flow behaviour. In particular, the analysis of the internal indicators based on active and reactive power (i.e., reactive acoustic power supplied to the core branch, active acoustic power loss through the feedback branch, amplification of the active acoustic power through the core branch, acoustic power delivered to the power extraction branch) explains the differences in the output power when the geometry of the compliance changes.

When comparing the predicted output power using the DeltaEC models and the experimental results, for the "Fba" and the "Fbb", the presence of circumferential clearance between the piston and the demonstrator resonator is not included in the model. Therefore, the resonator cylinder leaks, which implies a significant amount of flow through the piston–cylinder gap. To take into account this experimental effect, a reduction coefficient is employed to correct the theoretical data. This is attributed to the behaviour of the power extraction cylinder itself—the movement of the piston inside the cylinder becomes increasingly less cycle-repetitive when more energy is delivered to the power extraction branch—and to some discrepancies in modelling the actual behaviour of the power extraction component.

**Author Contributions:** Conceptualization, C.I. and J.V.; methodology, C.I. and J.G.; software, C.I. and J.G.; validation, J.V. and J.L.O.; formal analysis, C.I., J.G.; investigation, C.I., J.L.O., J.V. and J.G.; resources, J.L.O. and J.V.; data curation, C.I.; writing—original draft preparation, C.I; writing—review and editing, J.V. and J.L.O.; visualization, C.I. and J.V.; supervision, J.V. and J.L.O.; project administration, J.L.O. and J.V.; funding acquisition, C.I., J.V. and J.L.O. All authors have read and agreed to the published version of the manuscript.

**Funding:** This research was funded by the Comunidad de Madrid [grant SEGVAUTO 4.0-CM—P2018EEMT-4362]; and the Agencia Estatal de Investigación [grant RETOS 2018-RTI2018-095923-B-C22].

**Institutional Review Board Statement:** Not applicable.

**Informed Consent Statement:** Not applicable.

**Data Availability Statement:** The datasets generated during and/or analysed during the current study are available from the corresponding author on reasonable request.

**Acknowledgments:** The authors thank the Global Nebrija-Santander Chair of Energy Recovery in Surface Transport for their financial support.

**Conflicts of Interest:** The authors declare no conflict of interest.

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
