# Peer review of "Assessing the Performance of Design Variations of a Thermoacoustic Stirling Engine Combining Laboratory Tests and Model Results"

_machines, doi:10.3390/machines10100958_

Round 1

Reviewer 1 Report

In the “Introduction” should be briefly presented the thermoacoustic Stirling engine, which is actually the subject of the research carried out by the authors. This would contribute to a better understanding by readers of the research presented in the paper.

In row 15 "..active and Reactive Acoustic Power flow". The words "active" and "reactive" shouldn't both start with either the lowercase letter or the capital letter?

The title of Table 2 is on page 7 and the table on page 8.

The photos in Fig.4 are not clear enough. Resolution should be improved.

Figure 4 is on page 7 and fig.4. legend is on page 8.

Author Response

Point 1: In the “Introduction” should be briefly presented the thermoacoustic Stirling engine, which is actually the subject of the research carried out by the authors. This would contribute to a better understanding by readers of the research presented in the paper.

Response 1: The comment made by the reviewer on this point is appreciated. Following the reviewer suggestion the Figure 1 has been modified. Now the image shown in Figure 1a presents the thermoacoustic engine used in this study. In addition, lines 64 and 65 have been added to clarify that the description of the motor components presented is that of the engine used in this research. 

 Point 2: In row 15 "..active and Reactive Acoustic Power flow". The words "active" and "reactive" shouldn't both start with either the lowercase letter or the capital letter?

Response 2: It has been modified accordingly.

Point 3: The title of Table 2 is on page 7 and the table on page 8.

Response 3: This is now fixed.

Point 4: The photos in Fig.4 are not clear enough. Resolution should be improved.

Response 4: The figure is now displayed with better resolution.

Point 5: Figure 4 is on page 7 and fig.4. legend is on page 8.

Response 5: This is now fixed.

Reviewer 2 Report

The manuscript deals with very interesting issues concerning the evaluation of the efficiency of design variants of the thermo-acoustic Stirling engine. Heat recovery by internal combustion engine exhaust systems is a vast issue that is still undergoing modification. I consider the work an interesting look at this subject. However, I have a few minor comments:

1. The authors could write more about the project of which this work is a part;

2. Figure 1 should be described;

3. The description of the calculation model should contain more details;

4. Abbreviations used in the publication should be explained in parentheses.

The presented analysis of the results of the research conducted by the authors does not raise any objections. The work is transparent and insightful.

Author Response

Point 1: The authors could write more about the project of which this work is a part.

Response 1: Following the recommendation of the reviewer, a new paragraph has been added, from line 36 to line 43, to briefly describe the objective and scope of the RECUPERA project, which this study is part of.

Point 2: Figure 1 should be described

Response 2: We agree that it is important to describe this figure. It has been done accordingly.

Point 3: 3. The description of the calculation model should contain more details.

Response 3: Thank you for that comment. To provide more details of the model calculations, additional text has been added between lines 231 and 233 and also a paragraph has been added from line 244 to 252.  

Point 4: Abbreviations used in the publication should be explained in parentheses.

Response 4: This is now fixed.